

# Photocatalytic chloride to chlorine conversion by ionic iron in aqueous aerosols: A combined experimental, quantum chemical and chemical equilibrium model study

Marie Kathrine Mikkelsen[1], Jesper Baldtzer Liisberg[1], Maarten M. J. W. van Herpen[2], Kurt V. Mikkelsen[1], and Matthew S. Johnson[1]

[1]Department of Chemistry, H. C. Ørsted Institute, University of Copenhagen, Universitetsparken 5, DK-2100 Copenhagen Ø, Denmark
[2]Acacia Impact Innovation, Maarten van Herpen, Bernheze 5384 BB, The Netherlands;

**Correspondence:** Matthew S. Johnson (msj@chem.ku.dk)

**Abstract.**

Aerosol chamber experiments show that the ligand-to-metal charge transfer absorption in iron(III) chlorides can lead to the production of chlorine. Based on this mechanism, the photocatalytic oxidation of chloride in mineral dust-sea spray aerosols was recently shown to be the largest source of chlorine over the North Atlantic. However, there has not been a detailed analysis

of the mechanism including the aqueous formation equilibria and the absorption spectra of the iron chlorides; neither has there been an analysis of which iron chloride is the main chromophore. Here we present the results of experiments of photolysis FeCl$_3 \cdot$ 6H$_2$O in specific wavelength bands, an analysis of the absorption spectra of the title compounds from $n$=1..4 made using density functional theory, and the results of an aqueous phase model that predicts the abundance of the iron chlorides with changes in pH and ion concentrations. Transition state analysis is used to determine the energy thresholds of the dissociations

of the species. Based on a speciation model with conditions extending from dilute water droplet to acidic seawater droplet to brine to salty crust, and the absorption rates and dissociation thresholds, we find that FeCl$_2^+$ is the most important species for chlorine production for a wide range of conditions. The mechanism was found to be active in the range of 400 to 530 nm with a maximum around 440 nm. We conclude that iron chlorides will form in atmospheric aerosols from the combination of iron(III) cations with chloride and that they will be activated by sunlight, generating chlorine (Cl$_2$/Cl) from chloride (Cl$^-$), in a process

that is catalytic in both chlorine and iron.

## 1 Introduction

Common components of atmospheric mineral dust including TiO$_2$ and Fe$_2$O$_3$ are photocatalytically active (Ponczek and George, 2018) and yet evidence of this playing a meaningful role in the atmospheric radical budget has been elusive (Abou-

Ghanem et al., 2020; Chen et al., 2012). Recently, a large new source of chlorine atoms was discovered resulting from the





combination of Sahara dust with sea spray aerosol over the North Atlantic (van Herpen et al., 2023). The mechanism is triggered when Sahara dust mixes with sea spray aerosol in the marine boundary layer. Iron is released from the minerals forming iron chlorides which can absorb sunlight releasing a chlorine atom, which is emitted from the aerosol as molecular chlorine. The $Cl_2$ is then photolysed by sunlight yielding Cl in the gas phase (Wittmer et al., 2015, 2016; Wittmer and Zetzsch, 2017).

The chlorine produced by mineral dust-sea spray aerosols is estimated to produce 41 % of the chlorine over the Atlantic, impacting methane directly ($Cl + CH_4$) and indirectly (reduction in $[O_3]$ from $Cl + O_3$ reduces OH source). Oeste proposed a method for intentionally increasing the production of chlorine using iron salt aerosol to achieve atmospheric methane removal (AMR) (Oeste, 2009; Meyer-Oeste, 2014). The use of chlorine from any source as a climate intervention was recently evaluated by Li et al. (2023).

Traditionally, the tropospheric chlorine cycle (Saiz-Lopez and von Glasow, 2012; Simpson et al., 2015) begins with the formation of sea spray aerosol (Nielsen and Bilde, 2020), which are known to be ultrafine particles with high acidity (Angle et al., 2021). Acids such as $HNO_3$ and $H_2SO_4$ deposit forcing HCl into the gas phase which can react with OH to produce chlorine atoms, $HCl + OH \rightarrow Cl + H_2O$ (Young et al., 2014). Cl reacts with ozone, impacting the formation of hydroxyl radicals, and it reacts with methane and other hydrocarbons, reforming HCl (Chang and Allen, 2006; Knipping and Dabdub, 35  2003; Badia et al., 2019).

Chlorine production is poorly constrained, for example, Cl is estimated to remove between 0.8 and 3.3 % of tropospheric methane, depending on the model (Allan et al., 2007; Hossaini et al., 2016; Sherwen et al., 2016; Gromov et al., 2018; Li et al., 2022). Multiple lines of evidence show chlorine concentrations in the troposphere exceed what can be explained with existing mechanisms. These include 1. $^{13}C$ depletion in CO in air samples from Barbados (Mak et al., 2003), a signature of 40  methane oxidation by chlorine 2. Anomalies in the CO:ethane ratio seen at Cape Verde (Read et al., 2009). 3. Observations of elevations in the concentration of HOCl above what can be explained with standard chemistry (Lawler et al., 2011). 4. A comprehensive simulation of tropospheric chlorine using the GEOS-Chem global 3-D model of oxidant–aerosol–halogen atmospheric chemistry could not explain the elevated $Cl_2$ mixing ratios measured in the boundary layer in the WINTER aircraft campaign (Wang et al., 2019).

Additional chlorine production impacts our understanding of the methane budget because the abundance of $^{13}C$ in atmospheric methane is used to constrain emissions sources, and because $Cl + CH_4$ has a large kinetic isotope effect, while the main atmospheric methane sink reaction $OH + CH_4$ does not. The reaction of $CH_4$ with Cl has a carbon kinetic isotope effect (KIE) of $^{13C}KIE_{Cl} = 1.066 (\pm 0.002)$ at 297 K, which is around 17 times more fractionating than methane oxidation with OH radicals (Saueressig et al., 2001; Cantrell et al., 1990; Saueressig et al., 1995). The discovery of a new chlorine source means that 50  methane sources must be more depleted than had been recognized, leading to the conclusion that previous methane emissions budgets, which did not include the new chlorine source, likely underestimate biogenic methane (e.g. wetlands and agriculture), and overestimate the fossil fuel source (van Herpen et al., 2023). To understand the methane budget it is imperative to fully characterize the chlorine production mechanism and to see how it will vary with chemical conditions such as pH, chloride concentration and the concentrations of possibly interfering ions.





Historically, iron(III) chloride has been believed to form four complexes: $FeCl^{2+}$, $FeCl_2^+$, $FeCl_3$, and $FeCl_4^-$ (Gamlen and Jordan, 1953). Uchikoshi et al. (2022) presented a model of iron(III) chloride species, which shows the most plausible species to be: $FeCl_2^+$, $FeCl_3$, $FeCl_4^-$, and $FeCl_6^{3-}$. With the use of a theoretical mathematical decomposition model called the "Multivariate Curve Resolution Alternative Least Squares (MCRALS)" and a "5 complex model", they determined that the combination of Cl coordination numbers are 0, 2, 3, 4, and 6. This work indicates that $FeCl^{2+}$ will not be formed and the highest chlorinated

complex, forming at the highest chloride concentrations, will be $FeCl_6^{3-}$. The research by Uchikoshi et al. (2022) shows $FeCl_2^+$, $FeCl_3$, and $FeCl_4^-$ forming at a lower chlorine concentration than previously expected.

In this study, we present a detailed description of the photocatalytic oxidation of chloride to chlorine based on four iron(III) chloride complexes: $FeCl^{2+}$, $FeCl_2^+$, $FeCl_3$, and $FeCl_4^-$. A combination of modelling, quantum chemical calculations and laboratory experiments explains the formation constants of the iron chlorides under changing conditions of pH and chloride

concentration, their absorption rates under tropospheric sunlight, the quantum yield of absorption resulting from the energy threshold for photodissociation to yield a Cl radical, and the production of $Cl_2$ from irradiation of $FeCl_3 \cdot 6H_2O$.

## 1.1   The $Fe(III)Cl_n^{3-n}$ system

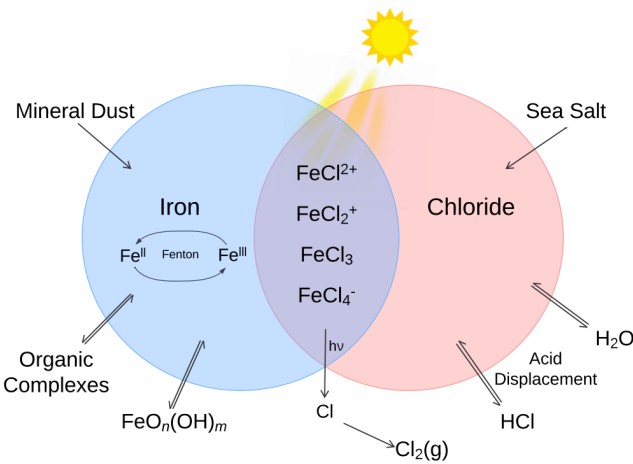

**Figure 1.** The primary sources and sinks for iron(II) and iron(III) ions and chloride in the atmospheric aerosols and their influence on the formation of iron(III) chlorides in a low pH environment.

Various Fe(III) complexes form in marine boundary layer aerosol, where the speciation depends on pH, salt composition, chloride concentration, water activity and ionic strength. The Pourbaix diagram of aqueous iron shows that dissolved iron will

be in the form of Fe(III) and not $Fe(OH)_3$, Fe(II) or $Fe(OH)_2$ given acidic and oxidising conditions (pH<3.6 and $E_H$>0.72 V) (Harnung and Johnson, 2012). Given chloride and Fe(III), the iron(III) chlorides will form.

The central $Fe(III)Cl_n^{3-n}$ reactions that occur in an aerosol or the marine boundary layer are shown in Reaction R1 for $n$=0..3 (Lindén et al., 1993).



$$FeCl_n^{3-n} + Cl^- \ \rightleftharpoons \ FeCl_{n+1}^{2-n} \tag{R1}$$

Iron(III) chloride formation may be inhibited by the presence of other ions such as sulfate or fluoride, or by organic compounds such as oxalate that bind with Fe(III).

    The photolysis of the iron(III) chlorides is shown in general form in Reaction R2 for $n \geq 1$. In this ligand to metal electron transfer absorption Fe(III) is reduced to Fe(II) and chloride is oxidised yielding a chlorine atom (Lindén et al., 1993; Nadtochenko and Kiwi, 1998).

$$FeCl_n^{3-n} + h\nu \ \rightarrow \ FeCl_{n-1}^{3-n} + Cl\cdot \tag{R2}$$

    For example for $n = 1$, reaction R2 has a quantum yield of $0.5 \pm 0.1$ at a wavelength of 347 nm. At higher pH the iron(III) chloride complexes compete with iron hydroxide complexes (also depending on $[Cl^-]$), Reaction 1 and 2.

$$Fe^{3+} + OH^- \ \rightleftharpoons \ FeOH^{2+} \tag{1}$$
$$FeOH^{2+} + OH^- \ \rightleftharpoons \ Fe(OH)_2^+ \tag{2}$$

Iron(III) hydroxide complexes can be photolysed, similar to iron(III) chlorides, however only some of the iron is photoreduced; Reaction 3 produces OH radicals with a quantum yield of $0.21 \pm 0.04$ at a wavelength of 347 nm (Nadtochenko and Kiwi, 1998). The UV absorption spectra of $Fe(OH)^{2+}$ and $Fe(OH)_2^+$ have been measured (Loures et al., 2013; Korte et al., 2011).

$$FeOH^{2+} + h\nu \ \rightarrow \ Fe^{2+} + OH\cdot \tag{3}$$
$$Fe(OH)_2^+ + h\nu \ \rightarrow \ FeOH^{2+} + OH^- \tag{4}$$

    In the aerosol process, the Fe(II) product is oxidised back to Fe(III) by one half of the Fenton process. In one example, for marine mineral aerosol, Zhu et al. found the oxidation rate to be $0.19\,\mathrm{min}^{-1}$ (Zhu et al., 1993). The Fenton process describes how Fe(II) and Fe(III) act as a catalyst pair, breaking down hydrogen peroxide and generating radicals (H. J. H. Fenton, 1894). The Fenton reactions are shown in Reactions 5 to 7. Hydroxyl may react or escape to the gas phase.

$$Fe^{2+} + H_2O_2 \ \rightarrow \ Fe^{3+} + OH\cdot + OH^- \tag{5}$$
$$Fe^{3+} + H_2O_2 \ \rightarrow \ Fe^{2+} + HO_2\cdot + H^+ \tag{6}$$
$$Net: 2H_2O_2 \ \rightarrow \ OH\cdot + HO_2\cdot + H_2O \tag{7}$$



Wittmer and Zetzsch (2017) proposed that the production of OH will enhance $Cl_2$ production. This route could be enhanced in chloride-rich environments. Several sources of OH are known including photocatalytic minerals (Chen et al., 2012) and Fenton degradation of $H_2O_2$ reaction 5.

$$OH^{\cdot} + Cl^- \rightarrow ClOH^{\cdot-} \tag{8}$$

$$ClOH^{\cdot-} + H^+ \rightarrow ClOH_2^{\cdot} \tag{9}$$

$$ClOH_2^{\cdot} \rightarrow Cl\cdot + H_2O \tag{10}$$

The reactions discussed above take place in the aqueous phase. In order for the chlorine atom produced in R1 to impact gas phase chemistry, it must escape the particle. The following Reactions 11 and 12 lead to the production of $Cl_2(aq)$.

$$Cl^- + Cl\cdot \rightleftharpoons Cl_2^- \tag{11}$$

$$2Cl_2^- \rightleftharpoons Cl_2 + 2Cl^- \tag{12}$$

$$Cl_2^- + Cl\cdot \rightleftharpoons Cl_2 + Cl^- \tag{13}$$

Chlorine atoms may be lost before escape as molecular chlorine in the gas phase. Possible mechanisms include failure of the chlorine atom produced by photolysis to escape the solvent cage or diffusion back into contact leading to reformation of the iron chloride, and reaction of chlorine with condensed phase hydrocarbons forming $HCl/Cl^-$.

Once in the gas phase molecular chlorine is activated by light, shown in Reaction 14. $Cl_2(g)$ absorbs in a band centred at 330 nm, Maric et al. (1993).

$$Cl_2(g) + h\nu \rightarrow 2Cl\cdot(g) \tag{14}$$



**Table 1.** Species and concentrations listed for each AEM scenario. Two different iron concentrations are used for each: iron seawater concentration, marked S and iron aerosol concentration, marked A.

| Aqueous Equilibrium Model (AEM) | | | |
|---|---|---|---|
| Models: | Simple | Sulphate | Seawater |
| Species | Concentration / mol/kg | | |
| Total Fe $_S^{(a)}$ | $9.76 \cdot 10^{-13}$ | $9.76 \cdot 10^{-13}$ | $9.76 \cdot 10^{-13}$ |
| $Fe_S^{2+}$ $^{(b)}$ | $7.32 \cdot 10^{-14}$ | $7.32 \cdot 10^{-14}$ | $7.32 \cdot 10^{-14}$ |
| $Fe_S^{3+}$ $^{(b)}$ | $9.02 \cdot 10^{-13}$ | $9.02 \cdot 10^{-13}$ | $9.02 \cdot 10^{-13}$ |
| Total Fe $_A^{(c)}$ | $9.17 \cdot 10^{-4}$ | $9.17 \cdot 10^{-4}$ | $9.17 \cdot 10^{-4}$ |
| $Fe_A^{2+}$ $^{(b)}$ | $6.88 \cdot 10^{-5}$ | $6.88 \cdot 10^{-5}$ | $6.88 \cdot 10^{-5}$ |
| $Fe_A^{3+}$ $^{(b)}$ | $8.48 \cdot 10^{-4}$ | $8.48 \cdot 10^{-4}$ | $8.48 \cdot 10^{-4}$ |
| $H^+$ $^{(d)}$ | 0 | 0 | 0 |
| $Cl^-$ | $5.47 \times 10^{-1}$ | $5.47 \times 10^{-1}$ | $5.47 \times 10^{-1}$ |
| $Na^+$ | $4.69 \times 10^{-1}$ | $4.69 \times 10^{-1}$ | $4.69 \times 10^{-1}$ |
| $SO_4^{2-}$ | | $2.83 \times 10^{-1}$ | $2.83 \times 10^{-1}$ |
| $Mg^{2+}$ | | | $5.24 \times 10^{-2}$ |
| $Ca^{2+}$ | | | $1.03 \times 10^{-2}$ |
| $K^+$ | | | $1.02 \times 10^{-2}$ |
| $Sr^{2+}$ | | | $9.12 \times 10^{-5}$ |
| $HCO_3^-$ | | | $2.33 \times 10^{-3}$ |
| $Br^-$ | | | $8.54 \times 10^{-4}$ |
| $F^-$ | | | $6.79 \times 10^{-5}$ |
| B | | | $4.13 \times 10^{-4}$ |
| $H_4SiO_4$ $^{(e)}$ | | | $3 \times 10^{-5}$ |

$^{(a)}$ The mean value of the total iron concentration in seawater was obtained from Achterberg et al. (2001). $^{(b)}$ Ratio of $Fe^{2+}$ and $Fe^{3+}$ is 7.5% and 92.5% respectively, from Zhu et al. (1993). $^{(c)}$ The total iron concentration in aerosols obtained from Hsu et al. (2010). $^{(d)}$ Required species for the AEM. $^{(e)}$ Value obtained from Harnung and Johnson (2012). Concentrations for unmarked species are found in Stumm and Morgan (2012).

## 2 Methods

This section will include method details of the study divided into three main parts: Aqueous Equilibrium Model (AEM), Ab Initio Calculations, and Laboratory Experiments.



## 2.1 Aqueous Equilibrium Model Methods

Visual MinteQ is a chemical equilibrium model that calculates the equilibrium speciation for the input species and predicts
their concentration (Gustafsson, 2014). The program has been used to evaluate species with direct or indirect effects on iron(III) induced chloride production.

Three AEM scenarios shown in Table 1, called Simple, Sulphate, and Seawater, are used. The species $FeCl^{2+}$, $FeCl_2^+$, and $FeCl_3$ were manually added to the database, shown in the Appendix Table A1 (Tagirov et al., 2000). Two different iron concentrations correlating to seawater and aerosol concentrations are used in all models, where the ratio between $Fe^{2+}$ and
$Fe^{3+}$ is estimated to be 7.5 to 92.5 % (Achterberg et al., 2001; Hsu et al., 2010; Zhu et al., 1993). The Simple model shows the relation between iron chloride species, the Sulphate model evaluates the potential effect of $SO_4^{2-}$ on iron chlorides, and the Seawater model includes all major ions found in seawater (Harnung and Johnson, 2012; Stumm and Morgan, 2012). The results from each of the three models is shown as species concentrations and iron species as a fraction of total iron, as a function of pH.

## 2.2 Ab Initio Calculations

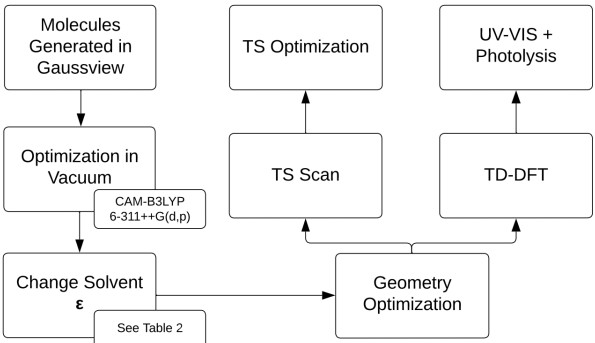

**Figure 2.** Overview of the computational strategy, highlighting key components: the relative permittivity ($\epsilon_r$), transition state (TS), and time-dependent density functional theory (TD-DFT).

Figure 2 describes the computational method, initialized by the generation of the molecules in Gaussview followed by geometry optimizations in vacuum (Frisch et al., 2016). The density functional theory method CAM-B3LYP/6-311++G(d,p) was used for all calculations (Yanai et al., 2004; Francl et al., 1982; McLean and Chandler, 1980; Spitznagel et al., 1987). Solvents were modelled using the PCM model (Tomasi et al., 1999). The relative permittivity, $\epsilon_r$, for the solvents is in Table 2.
See Appendix A2 for the geometries of the iron(III) chlorides.

A transition state scan was made to evaluate the energy when a chlorine atom leaves the system. The TS scan was used to estimate a TS barrier. In a few cases (mainly $FeCl_4^-$) in some solvents, the TS optimizations did not converge. TD-DFT



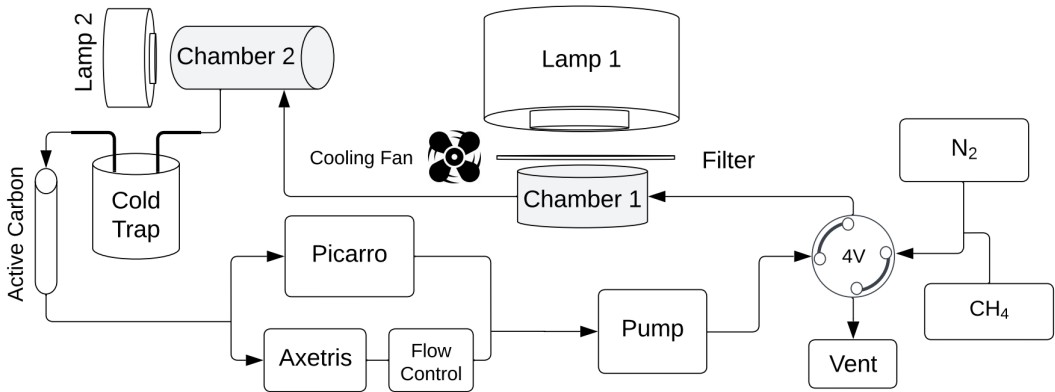

**Figure 3.** Overview of the system used in the laboratory experiments. The sample is located in "Chamber 1". The volume of Chamber 1 and 2 is 0.36 and 0.45 L, respectively. The four-port valve, "4V", changes the system between flow and loop mode.

**Table 2.** The set of relative permittivities (relative permittivity, $\epsilon_r$) used to represent various solvents with the PCM model.

| Solvent | $\epsilon_r$ |
|---|---|
| Water [a] | 78.4 |
| NaCl solutions [b] | 78.3 |
|  | 77.5 |
|  | 77.3 |
|  | 73.6 |
|  | 68.3 |
| Seawater [b] | 69.0 |
| Arbitrary investigation [c] | 60.0 |
|  | 50.0 |
|  | 40.0 |
|  | 30.0 |
|  | 20.0 |
| Solid FeCl$_3$, static [d] | 10.0 |
| Solid FeCl$_3$, optic [d] | 2.0 |

[a] Value obtained from Gaussview, Frisch et al. (2016). [b] Values obtained from Midi et al. (2014). [c] Arbitrary values were only used for FeCl$_3$. [d] Estimations.

calculations were used to explore the excitation energies of the molecules. Photolysis rates $j_A$ were calculated using Equation 15.





$$j_A = \int\limits_{\lambda_1}^{\lambda_2} \phi_A(\lambda,T) \cdot \sigma_A(\lambda,T) \cdot I(\lambda) d\lambda \tag{15}$$

where the variables represent the molecular absorption cross-section, $\sigma$ ($cm^{-2}$), spectral actinic flux, $I$ ($n_{hv}$ $cm^2$ $s^{-1}$ $nm^{-1}$), and quantum yield, $\phi$ (set to 1) (Seinfeld and Pandis, 2008). The absorption cross-section was evaluated computationally, and the spectral actinic flux was extracted with the TUV model in the Atlantic, west of Cape Verde (18.97°N, 39.12°W, date 18/07/2022, 12:00) (Madronich et al., 2002).

## 2.3 Experimental Method

The experimental system is shown in Figure 3, where gasses are introduced as shown at right towards a 4 port valve '4V' used to choose a flow-through or loop pattern. The average flow was measured to be 125 mL/min. A sample of $FeCl_3 \cdot 6H_2O$ is placed in Chamber 1 (volume 0.36 L) illuminated by a Xenon lamp denoted Lamp 1. For wavelength-controlled experiments a bandpass filter is placed on top of Chamber 1. Chamber 2 is a photolysis chamber with a volume of 0.45 L and a high-power UV LED light source, denoted "Lamp 2", that ensures chlorine emitted by the sample in Chamber 1 is photolysed. The cold trap and active carbon trap ensure that organic molecules and carbon dioxide are captured, which protects the instruments and removes possible interference. The airflow is divided for analysis by the Picarro "G2201-i Analyser for Isotopic $CO_2/CH_4$" and the Axetris "LGD Compact-A $CH_4$" which measure $CH_4$, $\delta^{13}CH_4$, $CO_2$, and $H_2O$. The flow controller is set to 125 mL/min and the pump ensures flow through the system.

In each experiment the active carbon trap and cold trap are flushed and a new sample of 20 mg $FeCl_3 \cdot H_2O$ in a 10 mL beaker is placed in a cleaned Chamber 1. Methane is introduced to achieve a nominal mole fraction of 95 ppm and the system is switched to loop mode. The concentrations of methane and carbon dioxide are monitored and the leak test of the system is performed after the concentrations have stabilized. System stability is monitored for 10 minutes, Lamp 1 is turned on for 15 minutes, and finally an additional 10 minutes of measurement verify system stability. The duration of an experiment is from 1.5 to 2.5 hours. Three repetitions were made for each bandpass filter. The experimental procedure is described in greater detail in Appendix Figure A1.

Two light sources were used: Lamp 1 was a Xenon lamp from Eimac, see Figure 4, and Lamp 2 was a Luminus SST-10-UV Surface Mount LED lamp, see Appendix Figure A2. The spectrum of Lamp 1 is shown with a dashed line in Figure 4; the light that enters Chamber 1 is illustrated with a solid line. The legend name refers to the centre wavelength of the bandpass filter in nm. The system and measurements of Chamber 1, bandpass filters, and Lamp 1 are illustrated in Appendix Figure A3.

## 3 Results

The results are presented in three sections: Aqueous Equilibrium Modelling, Ab Initio Calculations, and Laboratory Experiments.



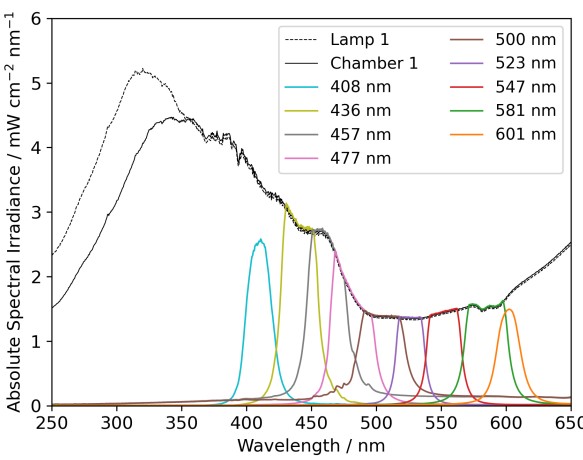

**Figure 4.** The measured absolute spectral irradiance of Lamp 1 before and after Chamber 1, and including bandpass filters. Recorded with the Ocean Optics spectrometer.

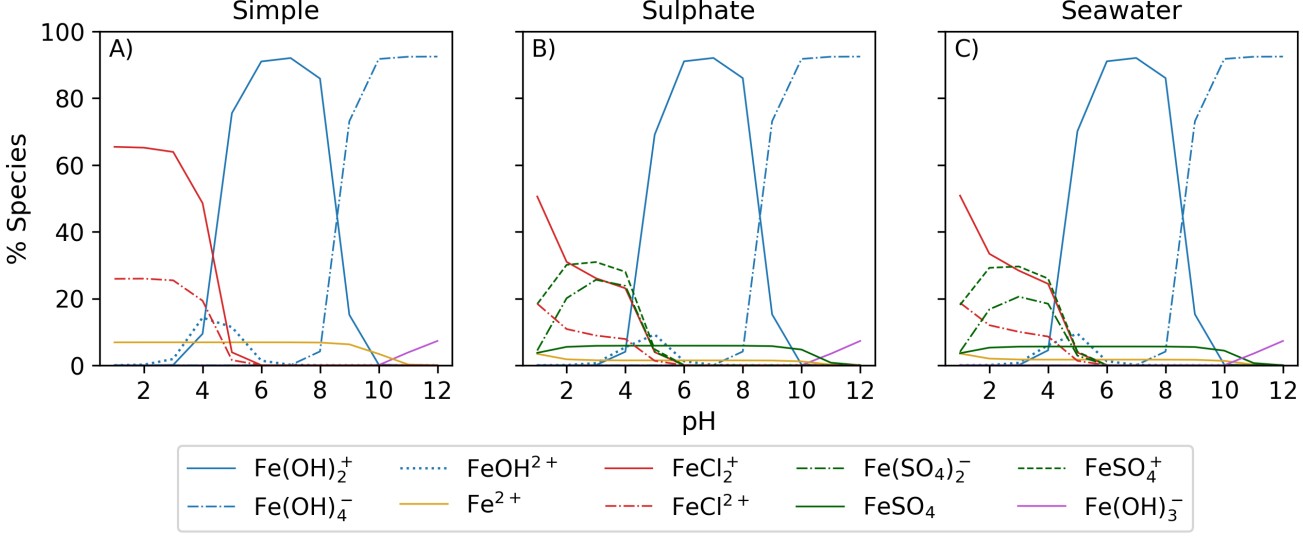

**Figure 5.** Three AEM scenarios with a seawater iron concentration of $9.76 \cdot 10^{-13}$ mol/kg. Only species above 5 % are shown.

## 3.1 Aqueous Equilibrium Modelling

170  Figure 5 displays the iron speciation from the three AEM scenarios as a function of pH, with a low iron concentration of 9.76 $\cdot 10^{-13}$ mol/kg according to seawater concentration. Iron(III) is stable at pH values less than 3.6. Above this the dominant





species are iron hydroxides. This trend is consistent for all models cf. Figure 5. According to the AEM scenarios, the two dominant iron(III) chloride species are $FeCl^{2+}$ and $FeCl_2^+$, both observed in a low pH environment. The Sulphate model, Figure 5B, shows that sulfate is able to block the availability of some iron to form iron chlorides. This changes the concentration by up to 40 % compared to the Simple model, Figure 5A. The seawater model, Figure 5C, does not show any major changes compared to the sulphate model. According to this study, sulphates are the main seawater anion competing with chloride for iron(III).

According to the AEM scenarios changing iron concentration from seawater (low) to aerosol (high) does not change iron speciation for the moieties shown in Figure 5. However, it has a small impact on the speciation of iron fluorides as seen in Appendix section B2, where all species above 0.1 % for all models are shown. The fraction of iron fluorides is below 3 % and they will not be discussed further.

The temperature dependence of $\alpha_{FeCl^{2+}}$ and $\alpha_{FeCl_2^+}$ was modelled with the Seawater scenario of the AEM illustrated in Figure 6 from 0 to 100 °C. The fraction of $FeCl^{2+}$ increases with increasing temperature and the opposite trend is seen for $FeCl_2^+$. With varying temperatures, a change of 70 and 45 % is found for $FeCl^{2+}$ and $FeCl_2^+$, respectively. Thus, temperature is an important parameter when calculating the rate of chlorine production from iron chlorides, as a change of 20 °C will significantly change iron speciation.

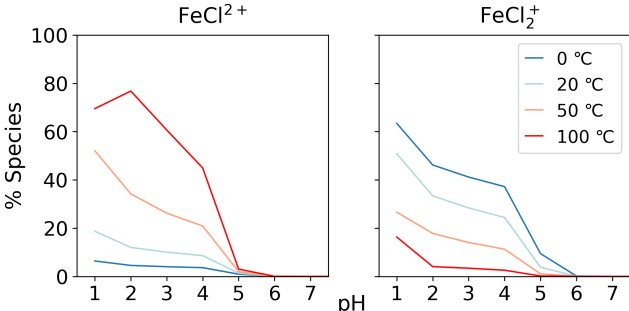

**Figure 6.** Temperature dependence of $\alpha_{FeCl^{2+}}$ and $\alpha_{FeCl_2^+}$ with seawater iron concentration of $9.76 \cdot 10^{-13}$ mol/kg ($Fe_S$).

## 3.2 Ab Initio Calculations

The four species $FeCl^{2+}$, $FeCl_2^+$, $FeCl_3$ and $FeCl_4^-$ were investigated computationally with the DFT functional CAM-B3LYP and basis set 6-311++G(d,p) (Yanai et al., 2004; Francl et al., 1982; McLean and Chandler, 1980; Spitznagel et al., 1987). UV-Vis spectra were extracted from TD-DFT calculations done with a variety of relative permittivities, shown in Table 2. UV-Vis spectra for $FeCl_3$ are displayed in Appendix Figure B1, resulting in an increasing red shift with increasing dielectric constant. This relative permittivity effect means that solvation of iron chlorides in water increases the absorption cross section at longer wavelengths where there is higher actinic flux, increasing the photolysis rate. The absorption spectra obtained for



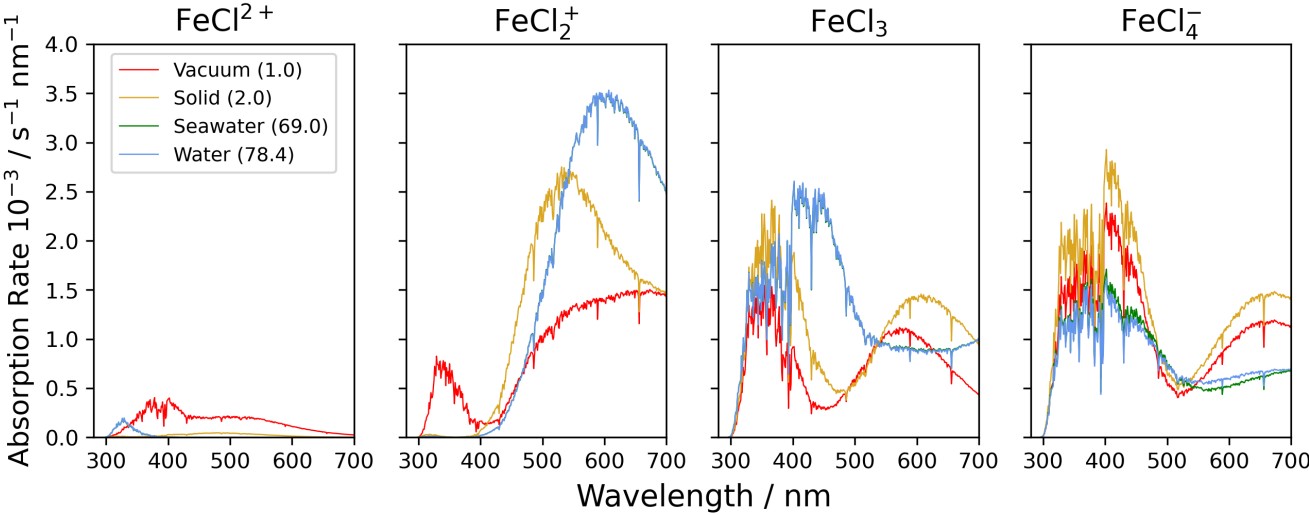

**Figure 7.** The absorption rate of $FeCl^{2+}$, $FeCl_2^+$, $FeCl_3$ and $FeCl_4^-$ in Vacuum (1.0), Solid $FeCl_3$ (2.0), Seawater (69.0), and Water (78.4). The value in parenthesis shows the relative permittivity of the solute. The absorption is calculated using the CAM-B3LYP/6-311++G(d,p) basis set. The spectral actinic flux is calculated using the TUV model over the Atlantic Ocean, west of Cape Verde (18.97°N, 39.12°W, date 18/07/2022, 12:00) (Madronich et al., 2002).

relative permittivities greater than 2 are seen to be virtually identical, thus, only the four environments vacuum, solid, seawater, and water, are discussed further.

Using the calculated UV-Vis spectra, the actinic flux and the quantum yield function, the photolysis spectra of $FeCl^{2+}$, $FeCl_2^+$, $FeCl_3$, and $FeCl_4^-$ are calculated, see Figure 7. Each of these species absorb at visible wavelengths, which has not been described previously. The modelled result for Seawater (green) is almost identical to the result for water (blue); the two spectra overlap in the figure. The lowest absorption rates are calculated for $FeCl^{2+}$ where all rates are below $0.5 \times 10^{-3}$ s$^{-1}$ nm$^{-1}$ for all four solvents. The AEM scenarios found $FeCl^{2+}$ to be the second most abundant iron(III) chloride species. Its role becomes less important however when the absorption spectrum is considered, in addition to the chromophore's concentration.

$FeCl_2^+$ in vacuum absorbs from 300 to 700 nm whereas solid, seawater, and aqueous $FeCl_2^+$ absorb in the range of 400 to 700 nm. $FeCl_2^+$ in water and seawater have the highest absorption rates of the investigated species, $3.5 \times 10^{-3}$ s$^{-1}$ nm$^{-1}$. The AEM scenarios found $FeCl_2^+$ to be the dominant iron(III) chloride species and when absorption spectrum is considered in combination with concentration, we conclude that $FeCl_2^+$ is the important chromophore for the catalytic, photo-oxidative conversion of chloride to chlorine in aqueous environments.

In the model, the absorption rates of $FeCl_3$ in seawater and water have a maximum of $2.6 \times 10^{-3}$ s$^{-1}$ nm$^{-1}$ at 402 nm. Similar trends are calculated for $FeCl_4^-$ where the solid has a maximum at 400 nm of $2.8 \times 10^{-3}$ s$^{-1}$ nm$^{-1}$. The absorptions in vacuum and solid have similar trends for both $FeCl_3$ and $FeCl_4^-$.





Visible light does not necessarily provide enough energy for the iron(III) chlorides to dissociate, and a transition state analysis can assist in the estimation of the energy thresholds. In Figure 8 the energy thresholds for photodissociation yielding a chloride radical are shown for the four iron(III) chloride species $FeCl^{2+}$, $FeCl_2^+$, $FeCl_3$, and $FeCl_4^-$. The photon energy is given in kJ/mol and converted to a photon wavelength in nm so it can be related to the solar spectrum. Near the surface the actinic flux spectrum becomes inconsequential at wavelengths shorter than 300 nm (Harnung and Johnson, 2012). This threshold is

shown in the figure with a yellow line.

     Figure 8 shows $FeCl^{2+}$ as the only species to have an energy threshold corresponding to a wavelength shorter than 300 nm for solid, seawater, and water and so this species can not be dissociated by near-surface solar excitation. According to the model, $FeCl_2^+$ has energy thresholds for solid, seawater, and water at 611, 603, and 605 nm, respectively. The highest absorption rates for $FeCl_2^+$ are in the region of 500 to 700 nm (see Figure 7), thus these thresholds very likely impact the photolysis rate.

Sunlight at the surface has photons with enough energy for the dissociation of $FeCl_2^+$.

     According to the transition state model, $FeCl_3$ is the only species that can be photolysed by all visible wavelengths and even into the near-infrared, for any of the solvents. For $FeCl_4^-$ the only sunlight-limiting energy threshold is in vacuum, at 573 nm. However, as $FeCl_4^-$ absorbs at wavelengths shorter than this threshold sunlight will dissociate this species under vacuum conditions.

General trends in Figure 8 show vacuum to be an outlier that either has a higher or lower energy threshold than solid, seawater, and water. Therefore, solvent effects play an important role when estimating the energy thresholds and absorption rates of the dissociation of these iron(III) chloride species.

     To relate the absorption rates and the energy dissociation threshold, the integrated absorption rates are listed in Table 3. The integrated absorption rate is integrated from 280 to 700 nm, as seen in Figure 7. The "cut-off" is integrated from 280 nm to the

dissociation energy threshold regarding each species, as seen in Figure 8. The deviation between "Full" and "Cut-off" is listed in percentages for each species as a measure of the cut-off impact.

     As listed in Table 3 $FeCl^{2+}$ has full absorption rates of 32, 4, 3, and 3 $s^{-1}$ in a vacuum, solid, seawater, and water, respectively. This species generally has the lowest integrated absorption rates which decrease significantly from vacuum to solid, seawater, and water. When the energy dissociation threshold is included, the integrated absorption rates for solid, seawater, and

water decrease to zero.

     $FeCl_2^+$ has full absorption rates of 177, 256, 312, and 312 $s^{-1}$ in vacuum, solid, seawater, and water, respectively. Including the cut-off significantly decreases the absorption rates by 45, 58 and 57 % for solid, seawater and water, respectively.

     $FeCl_3$ has full absorption rates of 150, 222, 259, and 259 $s^{-1}$ in vacuum, solid, seawater, and water, respectively. $FeCl_3$ is the only species that does not have sunlight-limiting energy thresholds; hence, the full and cut-off rates are identical. Furthermore,

$FeCl_3$ has a higher cut-off absorption rate for seawater and water than the other species, which increases the importance of this species.

     $FeCl_4^-$ has full absorption rates of 220, 272, 164, and 164 $s^{-1}$ in vacuum, solid, seawater, and water, respectively. The rates significantly decrease from vacuum to seawater and water. However, including the cut-off significantly decreases the rate in vacuum by 48 %, whereas the rate in water and seawater is unaffected and is consequently faster than the rate in vacuum.





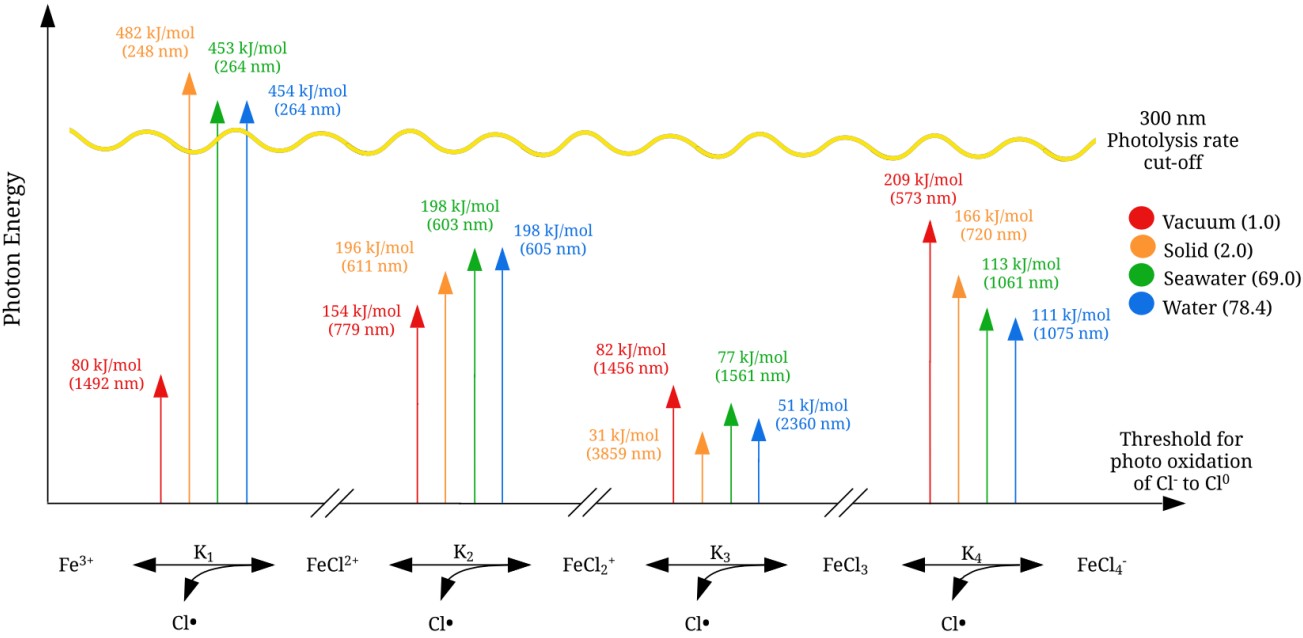

**Figure 8.** Energy thresholds for photodissociation yielding a chloride radial calculated with CAM B3LYP/6-311++G(d,p). The sunlight photolysis rate cut-off is illustrated in yellow. The relative permittivity for the solutes is given in parentheses.

Including the sunlight limiting energy thresholds, $FeCl_3$ is the species that has the fastest integrated absorption rates in seawater and water of 259 and 259 s$^{-1}$, respectively. Across all species, seawater and water have virtually identical rates, which further confirms that quantum calculations made in water can be used to evaluate behaviour in seawater.

### 3.3   Laboratory Study

The change in measured methane concentration was used as a proxy for chlorine radical production. The $x$-axis in Figure 9 is
ordered according to the centre wavelength of the bandpass filter used in the experiments, where "none" indicates that no filter was used. The spectral irradiance for the filters is displayed in Figure 4.

The absolute and energy normalized methane removal is shown in Figure 9 (Left) in units of $CH_4$ molecules per $FeCl_3 \cdot 6H_2O$ molecule. The energy normalization was calculated with respect to the absolute removal at 408 nm and the integrated spectral irradiance for each bandpass filter.

The largest average methane removal occurs at 436 nm. At this wavelength the removal corresponds to $3.3 \times 10^{-3}$ and $2.0 \times 10^{-3}$ $CH_4$ per $FeCl_3 \cdot 6H_2O$ for absolute and energy normalized removal, respectively. This means that chlorine generation is small relative to the iron chloride reservoir in the sample. The lowest average methane removal occurs at 523 nm, $0.3 \times 10^{-3}$ and $0.8 \times 10^{-3}$ $CH_4$ per $FeCl_3 \cdot 6H_2O$ for absolute and energy normalized removal, respectively.





**Table 3.** Integrated Absorption Rates, s$^{-1}$. "Full" is integrated from 280 to 700 nm, as displayed in Figure 7. "Cut-off" is integrated from 280 nm to the dissociation energy threshold in Figure 8. The deviation is calculated between the Full and the Cut-off absorption rates.

| | | FeCl$^{2+}$ | FeCl$_2^+$ | FeCl$_3$ | FeCl$_4^-$ |
|---|---|---|---|---|---|
| **Vacuum** | Full / s$^{-1}$ | 32 | 177 | 150 | 220 |
| | Cut-off / s$^{-1}$ | 32 | 177 | 150 | 113 |
| | Deviation / % | 0 | 0 | 0 | 48 |
| **Solid** | Full / s$^{-1}$ | 4 | 256 | 222 | 272 |
| | Cut-off / s$^{-1}$ | 0 | 141 | 222 | 272 |
| | Deviation / % | 100 | 45 | 0 | 0 |
| **Seawater** | Full / s$^{-1}$ | 3 | 312 | 259 | 164 |
| | Cut-off / s$^{-1}$ | 0 | 131 | 259 | 164 |
| | Deviation / % | 100 | 58 | 0 | 0 |
| **Water** | Full / s$^{-1}$ | 3 | 312 | 259 | 164 |
| | Cut-off / s$^{-1}$ | 0 | 134 | 259 | 164 |
| | Deviation / % | 100 | 57 | 0 | 0 |

The reaction of methane with chlorine is highly fractionating, and an increase in $\delta^{13}CH_4$ would suggest that chlorine is the oxidizing agent and not a different oxidation pathway, e.g. hydroxyl radicals. The abundance of $^{13}C$ in $CH_4$, measured as $\delta^{13}C\text{-}CH_4$, is measured during the experiments with the Picarro G2201-i, and is shown in Figure 9 (Right) as the change in the delta value.

A chlorine steady-state (Cl SS) concentration was calculated from the observed $CH_4$ removal (seen in Figure 9 (Left)) to derive the concentration of chlorine required to remove an amount of methane (the Cl SS calculations are shown in Appendix section A6). The expected $\delta^{13}C\text{-}CH_4$ was calculated using the SS Cl concentration with the rate constants for the chlorine oxidation of $^{12}CH_4$ and $^{13}CH_4$. The expected $\delta^{13}C\text{-}CH_4$ value is shown in green together with the change in the delta value in Figure 9 (Right).

Oxidation of methane by OH radicals results in a fractionation 17 times smaller than for the methane - Cl reaction (Saueressig et al., 2001, 1995). A lower observed than expected change in the delta value would indicate an additional removal pathway with a lower kinetic isotope effect (e.g. OH). The observed and expected $\Delta\delta^{13}CH_4$ values are equal to within the experimental uncertainties, except for the experiment made using the 436 nm bandpass filter which is just outside the mutual uncertainties; we think this was an anomaly due to a calibration offset. Figure 9 (Right) shows a clear correlation between the expected and the observed $\delta^{13}CH_4$ indicating that chlorine is the oxidizing agent throughout all experiments.



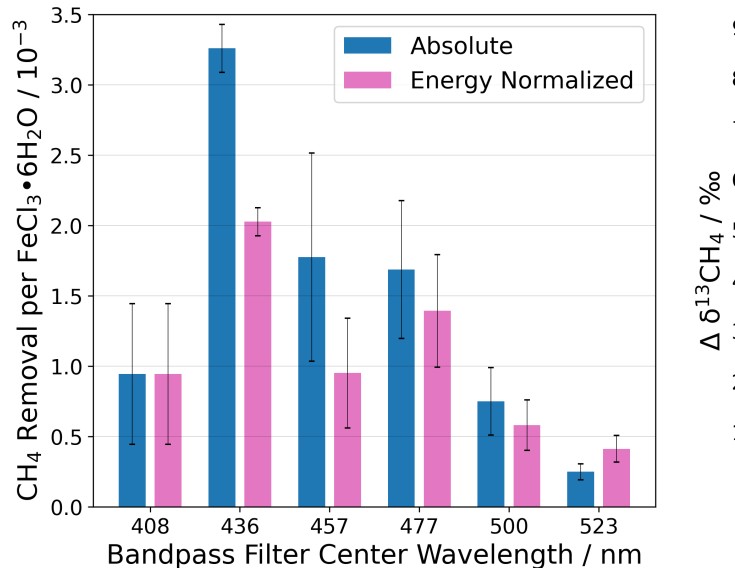
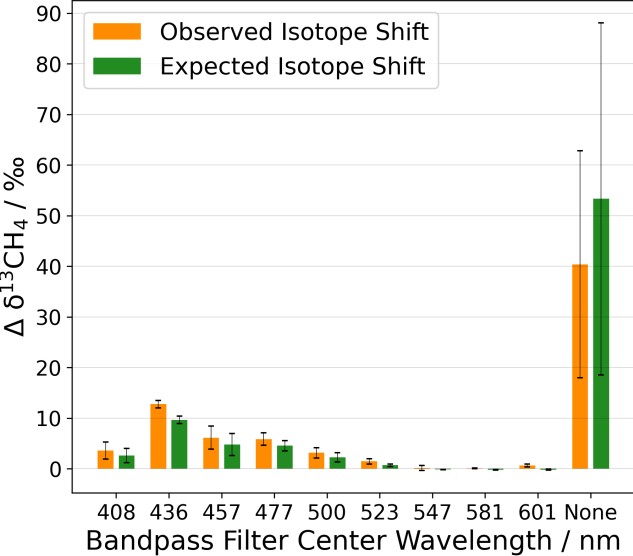

**Figure 9.** (Left) Chlorine generation measured as methane removal for photoexcitation in a series of wavelength intervals defined by bandpass filters. Without a filter the absolute $CH_4$ removal was $10.4 \times 10^{-3}$ $CH_4$ molecules per $FeCl_3 \cdot 6H_2O$ molecule. (Right) The change in the abundance of $^{13}C$ is shown as $\Delta\delta^{13}C - CH_4$. "none" indicates that no bandpass filter was used during the experiment. Steady state calculations of Cl are shown in Appendix section A6.

## 4 Conclusions

In this work, we have presented a detailed description of the photolysis of iron(III) chlorides in aerosols. Modelling, quantum chemical calculations and laboratory experiments have provided crucial insight into how iron(III) chloride chromophores behave in aqueous solutions under varying conditions of pH and $[Cl^-]$.

Three AEM scenarios were built to describe the speciation of the iron chlorides and the competition inhibiting their formation from hydroxide, sulphate, and other seawater ions. The results show that iron(III) chlorides are the dominant forms of iron at 280 pH<4.0, and iron(III) hydroxides at pH above this. The addition of sulphate changed the abundance of $FeCl_2^+$ by up to 40 %. The other seawater species we tested did not have a significant effect. The speciation of $FeCl^{2+}$ and $FeCl_2^+$ changes by up to 70 % over the temperature range of 0 to 100 °C.

Quantum chemical calculations made using the density functional theory method CAM-B3LYP/6-311++G(d,p) were used to investigate $FeCl^{2+}$, $FeCl_2^+$, $FeCl_3$, and $FeCl_4^-$. Solvent effects for vacuum, solid, seawater, and water were included using 285 the PCM model and relative permittivities. These calculations showed all four species absorb light in the visible spectrum. The seawater and water absorptions are predicted to be virtually identical. Dissociation energy thresholds were found using transition state calculations to evaluate the energy when a chlorine atom leaves the system. For $FeCl^{2+}$, $FeCl_2^+$, and $FeCl_4^-$ inclusion of the energy threshold significantly decreased the absorption rates. $FeCl_3$ was the only species without sunlight-



limiting energy thresholds in any solvent. $FeCl_3$ has the fastest integrated absorption rate of $259$ s$^{-1}$ and $FeCl^{2+}$ has the

slowest rate of $0$ s$^{-1}$ both in water.

We examined the photolysis of solid $FeCl_3 \cdot 6H_2O$ in nine wavelength intervals using bandpass filters with centre wavelengths ranging from 408 to 601 nm. The maximum methane removal was observed for the 436 nm bandpass filter to be $2.0 \times 10^{-3}$ $CH_4$ molecules per $FeCl_3 \cdot 6H_2O$, whereas the removal efficiency without a bandpass filter was $10.4 \times 10^{-3}$ $CH_4$ molecules per $FeCl_3 \cdot 6H_2O$ including energy normalization. No statistically significant methane removal was observed at wavelengths longer

than 523 nm. Based on the observed $\Delta \delta^{13}CH_4$, methane is oxidized by chlorine and not OH. Maximum chlorine production was found at an excitation wavelength around 436 nm.

The peak wavelength for $Cl_2$ production in the experiment was found at the same position as predicted by quantum chemistry. Thus the experiments confirm the prediction from quantum chemistry that visible light, abundant in the troposphere, is able to drive the iron catalysed photooxidation of chloride to chlorine.

Based on these results, we conclude that the catalytic efficiency of the iron chloride mechanism will be constrained by a number of environmental variables. The pH of the system clearly plays an important role, affecting the abundance of the $FeCl_n^{3-n}$ chromophores. Fenton oxidation of Fe(II) to Fe(III) will result in production of base, which is countered by the absorption of environmental acids like HCl, $H_2SO_4$, $HNO_3$ and organic acids. The concentration of chloride also impacts the abundance and distribution of the $FeCl_n^{3-n}$ chromophores and will be affected by equilibrium with atmospheric humidity,

and the emission and re-absorption of HCl. The presence of anions other than $Cl^-$, such as sulfate and fluoride, may reduce the availability of Fe(III) to make chloride chromophores. While the relative concentrations of ions in seawater are nearly constant throughout the oceans, dissolved organic matter (DOM) is complex and variable (Harnung and Johnson, 2012). The interation of DOM with iron and the iron-salt mechanism should be the subject of future research. In this work we have used a transition state model to determine the threshold frequency for photolysis. The dynamics of the process are more complicated

both prior to dissociation and after bond breaking, and both of these processes will affect the observed quantum yield. Finally, the probability of escape of oxidised chlorine as molecular chlorine will be impacted by the amount of condensed water and the concentrations of species in that water that react with chlorine radicals. Further research is needed in order to explore the catalytic efficiency of ionic iron, and the catalytic efficiency of chlorine recycling, under the many different conditions found in the troposphere. The utility of the iron chloride mechanism as a potential climate intervention will depend on these variables

and in addition, the atmospheric conditions under which the Cl is released. As shown by Li et al., paradoxically, additional Cl can cause an increase in methane lifetime under some common conditions in the atmosphere (Li et al., 2022).

## Appendix A: Supplementary Methods

### A1    Aqueous Equilibrium Model: Species added manually to the Visual MinteQ database





**Table A1.** Thermodynamic data for species added to the MinteQ models. The values were calculated from data by Tagirov et al. (2000) (Tagirov et al., 2000).

| Species | $M_w$ / (g mol$^{-1}$) | log(K) | $\Delta H$ / (kJ mol$^{-1}$) |
|---------|---------|--------|-------|
| $FeCl^{2+}$ | 91.3 | 2.6 | 43.5 |
| $FeCl_2^+$ | 126.8 | 3.9 | 1.3 |
| $FeCl_3$ | 162.2 | -2.7 | 146.8 |

## A2 Geometries of iron(III) chlorides

**Table A2.** Geometries of minima and TS for $FeCl^{2+}$. TS and relative permittivities are given in parentheses. The TS is not optimized if not given otherwise. Lennard-Jones potential. Units in Å.

| Atom | X | Y | Z |
|------|---|---|---|
| | | $FeCl^{2+}$ - Vacuum | |
| Fe | 0.00000000 | 0.00000000 | 0.89243200 |
| Cl | 0.00000000 | 0.00000000 | -1.36489500 |
| | $FeCl^{2+}$ - Vacuum (Optimized LJ Potential) | | |
| Fe | 0.00000000 | 0.00000000 | 1.36685000 |
| Cl | 0.00000000 | 0.00000000 | -2.09047700 |
| | | $FeCl^{2+}$ - Solid (2.0) | |
| Fe | 0.00000000 | 0.00000000 | 0.92908200 |
| Cl | 0.00000000 | 0.00000000 | -1.42094900 |
| | | $FeCl^{2+}$ - Solid (10.0) | |
| Fe | 0.00000000 | 0.00000000 | 0.97379200 |
| Cl | 0.00000000 | 0.00000000 | -1.48932800 |
| | | $FeCl^{2+}$ - Water (78.4) | |
| Fe | 0.00000000 | 0.00000000 | 0.98773100 |
| Cl | 0.00000000 | 0.00000000 | -1.51064700 |
| | $FeCl^{2+}$ - Water (Optimized TS, 78.4) | | |
| Fe | 0.00000000 | 0.00000000 | 1.25047300 |
| Cl | 0.00000000 | 0.00000000 | -1.97410000 |



**Table A3.** Geometries of minima and TS for $FeCl_2^+$. TS and relative permittivities are given in parentheses. The TS is not optimized if not otherwise indicated. Lennard-Jones potential. Units in Å.

| Atom | X | Y | Z |
|---|---|---|---|
| $FeCl_2^+$ - Vacuum | | | |
| Fe | 0.00000000 | 0.00000000 | 0.64324200 |
| Cl | 0.00000000 | 1.67657400 | -0.49189100 |
| Cl | 0.00000000 | -1.67657400 | -0.49189100 |
| $FeCl_2^+$ - Vacuum (LJ Potential) | | | |
| Fe | 0.00000000 | -1.14903500 | 0.00000000 |
| Cl | 0.93891500 | 4.80205800 | 0.00000000 |
| Cl | -0.93891500 | -3.04471000 | 0.00000000 |
| $FeCl_2^+$ - Solid (2.0) | | | |
| Fe | 0.00000000 | 0.00000000 | 0.77985900 |
| Cl | 0.00000000 | 1.54863200 | -0.59636300 |
| Cl | 0.00000000 | -1.54863200 | -0.59636300 |
| $FeCl_2^+$ - Solid (10.0) | | | |
| Fe | 0.00000000 | 0.00000000 | 0.82053800 |
| Cl | 0.00000000 | 1.50744300 | -0.62747000 |
| Cl | 0.00000000 | -1.50744300 | -0.62747000 |
| $FeCl_2^+$ - Water (78.4) | | | |
| Fe | 0.00000000 | 0.00000000 | 0.82832900 |
| Cl | 0.00000000 | 1.51047300 | -0.63342800 |
| Cl | 0.00000000 | -1.51047300 | -0.63342800 |
| $FeCl_2^+$ - Water (LJ Potential, 78.4) | | | |
| Fe | 0.00000000 | -1.53061900 | 0.00000000 |
| Cl | 1.53198900 | 2.27452700 | 0.00000000 |
| Cl | -1.53198900 | 0.06642000 | 0.00000000 |





**Table A4.** Geometries of minima and TS for FeCl$_3$. TS and relative permittivities are given in parentheses. The TS is not optimized if not otherwise indicated. Units in Å.

| Atom | X | Y | Z |
|------|-----------|-----------|------------|
| \multicolumn | FeCl$_3$ - Vacuum | | |
| Fe | 0.00000000 | 0.00000000 | 0.06229900 |
| Cl | 0.00000000 | 0.00000000 | 2.12724600 |
| Cl | 0.00000000 | 1.70272000 | -1.11126300 |
| Cl | 0.00000000 | -1.70272000 | -1.11126300 |
| | FeCl$_3$ - Vacuum (Optimized TS) | | |
| Fe | 0.00000000 | 0.19936900 | 0.00000000 |
| Cl | -0.23316300 | 2.26248200 | 0.00000000 |
| Cl | -1.59583200 | -1.78955500 | 0.00000000 |
| Cl | 1.82899600 | -0.77784400 | 0.00000000 |
| | FeCl$_3$ - Solid (2.0) | | |
| Fe | 0.00000000 | 0.00000000 | 0.09462900 |
| Cl | 0.00000000 | 0.00000000 | 2.17353500 |
| Cl | 0.00000000 | 1.66523400 | -1.15913100 |
| Cl | 0.00000000 | -1.66523400 | -1.15913100 |
| | FeCl$_3$ - Solid (10.0) | | |
| Fe | 0.00000000 | 0.00000000 | 0.16164100 |
| Cl | 0.00000000 | 0.00000000 | 2.27392600 |
| Cl | 0.00000000 | 1.58394800 | -1.26057100 |
| Cl | 0.00000000 | -1.58394800 | -1.26057100 |
| | FeCl$_3$ - Water (78.4) | | |
| Fe | 0.00000000 | 0.00000000 | 0.17565400 |
| Cl | 0.00000000 | 0.00000000 | 2.30557800 |
| Cl | 0.00000000 | 1.57451400 | -1.28711300 |
| Cl | 0.00000000 | -1.57451400 | -1.28711300 |
| | FeCl$_3$ - Water (Optimized TS, 78.4) | | |
| Fe | 0.26266300 | 0.28139600 | -0.18588400 |
| Cl | 2.45315100 | 0.14460200 | 0.13374700 |
| Cl | -1.91616200 | 1.03074500 | 0.11469600 |
| Cl | -0.93871000 | -1.60571700 | 0.03584900 |





**Table A5.** Geometries of minima and TS for $FeCl_4^-$. TS and relative permittivities are given in parentheses. The TS is not optimized if not otherwise indicated. Units in Å.

| Atom | X | Y | Z |
|---|---|---|---|
| $FeCl_4^-$ - Vacuum | | | |
| Fe | 0.0150980 | 0.0027010 | 0.0786570 |
| Cl | -1.8497650 | -0.0129990 | 1.2143290 |
| Cl | -0.2089620 | -1.6349200 | -1.3253390 |
| Cl | 0.0482380 | 1.8732870 | -1.0241150 |
| Cl | 1.9873980 | -0.2295000 | 1.0148250 |
| $FeCl_4^-$ - Vacuum (TS) | | | |
| Fe | -0.3705550 | 0.0003460 | -0.0214210 |
| Cl | 3.5129100 | -0.0002720 | -0.0031090 |
| Cl | -0.9707510 | -0.0127400 | 2.0163570 |
| Cl | -0.9874020 | 1.7870870 | -0.9795580 |
| Cl | -0.9880270 | -1.7746060 | -1.0009290 |
| $FeCl_4^-$ - Solid (2.0) | | | |
| Fe | -0.0121400 | -0.0032990 | 0.0983290 |
| Cl | 0.1515470 | 1.6578120 | -1.2840190 |
| Cl | -1.9913620 | 0.1824750 | 1.0231410 |
| Cl | -0.0442120 | -1.8511330 | -1.0394650 |
| Cl | 1.9025950 | 0.0158910 | 1.1499570 |
| $FeCl_4^-$ - Solid (10.0) | | | |
| Fe | -0.0039350 | -0.0073460 | -0.1784210 |
| Cl | 2.0378400 | 0.0695040 | -0.9607470 |
| Cl | 0.0120240 | -1.8310140 | 1.0092500 |
| Cl | 0.0139720 | 1.7030320 | 1.1614890 |
| Cl | -2.0578180 | 0.0697140 | -0.9371140 |
| $FeCl_4^-$ - Water (78.4) | | | |
| Fe | -0.0006960 | -0.0067460 | -0.2094960 |
| Cl | 2.0784560 | 0.0677370 | -0.8820680 |
| Cl | 0.0065630 | -1.8373510 | 0.9761920 |
| Cl | -0.0033190 | 1.7208020 | 1.1060030 |
| Cl | -2.0806360 | 0.0591290 | -0.8797220 |
| $FeCl_4^-$ - Water (TS, 78.4) | | | |
| Fe | -0.3705550 | 0.0003460 | -0.0214210 |
| Cl | 3.5129100 | -0.0002720 | -0.0031090 |
| Cl | -0.9707510 | -0.0127400 | 2.0163570 |
| Cl | -0.9874020 | 1.7870870 | -0.9795580 |
| Cl | -0.9880270 | -1.7746060 | -1.0009290 |





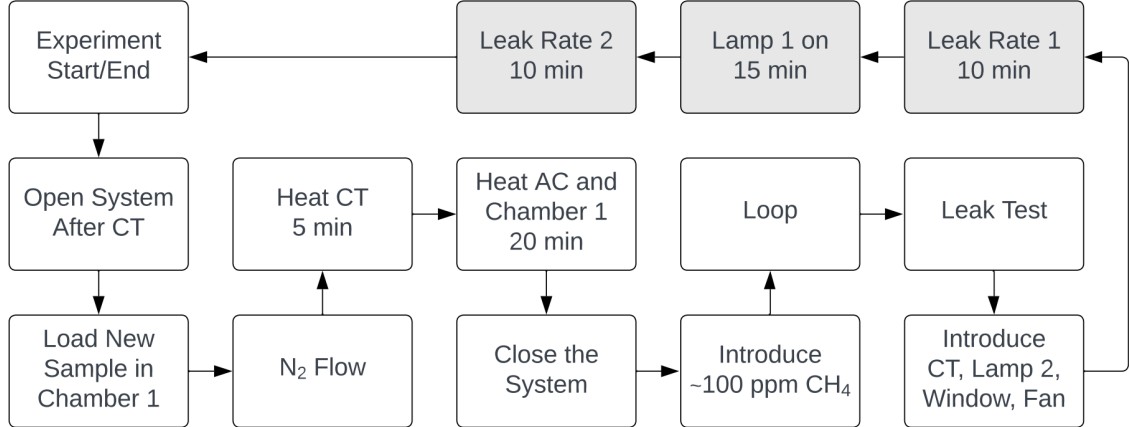

**Figure A1.** Overview of the experimental procedure. Data is retrieved during the three grey highlighted steps. Abbreviations: Cold Trap (CT), Active Carbon Trap (AC), different bandpass filters.

## A3    Experimental Procedure

When the experimental procedure begins, the system is opened between the active carbon trap and the two measuring instruments. While the system is open, the flow is increased, and an overpressure vent is closed to flush the system. A $FeCl_3 \cdot H_2O$ sample of 20 mg is now measured in a 10 mL beaker on a fine scale. Chamber 1 is cleaned with Milli-Q water on some paper and wiped again with dry paper. The beaker is placed in Chamber 1, and the chamber is closed. Two O-rings ensure a tight
closure of Chamber 1. While the system is open, the cold trap is heated for 5 minutes to flush condensed species out of the system. The active carbon trap is heated for 20 minutes to release the captured $CO_2$. Chamber 1 is heated simultaneously with the active carbon trap to minimize the temperature difference in Chamber 1 when Lamp 1 is turned on. After the heating processes, the flow is decreased by opening the over-pressure vent, and the system is closed. There is now flow through the whole system. During the process of closing the system, the $CO_2$ and $H_2O$ concentrations are recorded by the Picarro to ensure that the active
carbon trap works properly. Methane is now introduced with a syringe to the nitrogen flow. The methane concentration in the system rises to 200 to 400 ppm, and the four-port valve is closed at around 150 ppm. After the looping process is started the methane concentration is monitored to ensure it stabilizes at around 95 ppm. After approximately five minutes, the stability test begins. Due to a small underpressure in the system, the presence of a possible leak is quantified by spraying pure methane from a syringe at joints, seals and so on, and observing whether the methane concentration in the system increases. After the system
has passed the stability test the cold trap, filter, fan, and lamp 2 are initiated. The two instruments are both slightly sensitive to temperature, 'leak rate 1' is therefore initiated five minutes after the cold trap is introduced, to measure stability. The next step is recording the leak rate over 10 minutes before illumination. Lamp 1 is turned on for exactly 15 minutes, and the second leak rate is measured over the following 10 minutes. One experiment has now ended, and the procedure is repeated with a new



sample of $FeCl_3 \cdot H_2O$. The different bandpass filters and light sources are discussed in methods. Each of the 10 filters was used

for three experiments.





## A4 Light sources used in the experiments

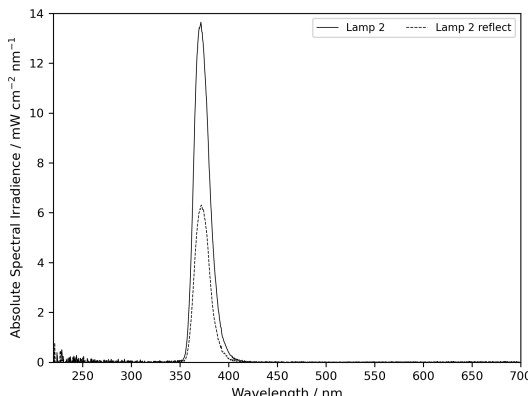

**Figure A2.** Light source spectra. Lamp 2 was a Luminus SST-10-UV Surface-Mount LED lamp. The spectrum is recorded by an Ocean Optics Flame-S Miniature UV-Vis-NIR spectrometer, and a fibre optic cable with a diameter of 600 $\mu$m before and after the chamber.

Lamp 1, bandpass filters, and chamber filters were placed according to the experimental setup when the irradiance was measured. The chamber setup with measurements can be seen in Figure A3. The Ocean Optics spectrometer was placed at a distance according to the bottom of the chamber, to estimate the light exposure for the sample. A calibration was used for the
Ocean Optics spectrometer using the standard manual and light source from Ocean Optics (DH-2000, UV-Vis-NIR calibration light source). The irradiance of Lamp 2 is shown in Figure A2.





## A5 Chamber 1 setup and measurements

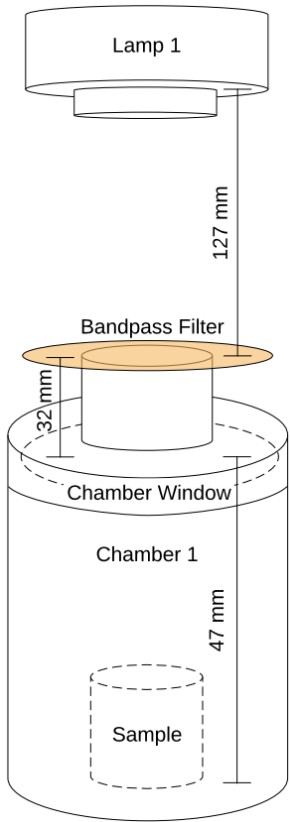

**Figure A3.** Results from experiments using the Chamber 1 setup during the experiments including the Ocean Optics spectrometer. Chamber 1 had a volume of 0.36 L.





## A6 Calculation of Steady State chlorine atom concentration and nominal $\delta^{13}$C-CH$_4$

The steady state concentration of chlorine atoms is calculated with the assumption of a constant source and sink of chlorine.

The decay of methane can be described as a pseudo-first-order reaction, in Equation A1:

$$CH_4(t) = CH_4(0) \times e^{-rt} \tag{A1}$$

Where $t$ is the time and $r$ is the first order loss rate in Equation A2. $k(^{12}CH_4)$ is the rate constant of the $^{12}CH_4$ + Cl reaction, and $[Cl]_{ss}$ is the chlorine concentration.

$$r = k(^{12}CH_4) \times [Cl]_{ss} \tag{A2}$$

The Cl SS concentration is isolated and used to calculate the expected isotope shift. The kinetic isotope effect $\alpha$ is used to define the isotopic fractionation $\epsilon$ (Johnson et al., 2002):

$$\alpha = \frac{k(^{12}CH_4)}{k(^{13}CH_4)} \tag{A3}$$

$$\epsilon = \alpha - 1 \tag{A4}$$

The final isotopic enrichment $\delta$ is calculated based on the enrichment of the starting material $\delta_0$, $\epsilon$, and the extent of reaction,
$f$:

$$\delta = \delta_0 + \epsilon \cdot ln(f) \tag{A5}$$





## Appendix B: Supplementary Results

### B1  Ab Initio Calculations: FeCl₃

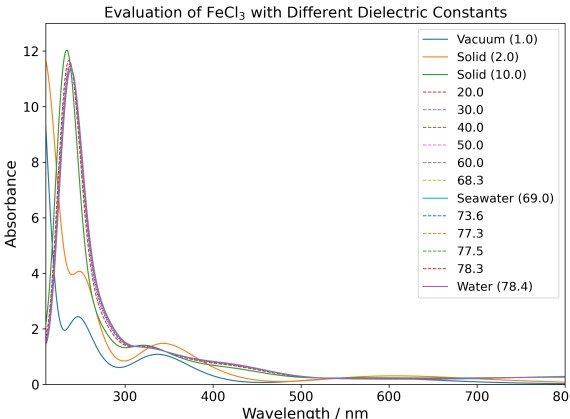

**Figure B1.** Absorption of FeCl₃ calculated with CAM-B3LYP/6-311++G(d,p). Solid (2.0) and solid (10.0) are estimated to represent the optic and static relative permittivity for the solid state of FeCl₃, respectively.

### B2  All Aqueous Equilibrium Models





**Figure B2.** Simple, Sulphate and Seawater Aqueous Equilibrium Models with a total iron concentration of $9.76\times10^{-13}$ mol/kg (Fe$_S$). The right column is zoomed in from 0 to 5 %.



**Figure B3.** Simple, Sulphate and Seawater Aqueous Equilibrium Models with a total iron concentration of $9.17 \times 10^{-4}$ mol/kg ($Fe_A$). The right column is zoomed in from 0 to 5 %.



*Data availability.*  Contact Matthew S. Johnson, msj@chem.ku.dk, for available data sets.

*Author contributions.*  MKM, MvH and MSJ designed the research. MKM and JBL prepared the laboratory experiments with advise from MvH and MSJ. MKM performed and analysed the quantum chemistry calculations, modelling results, and laboratory experiments. JBL helped with the laboratory experiments and the data analysis was carried out by MKM with input from JBL, MSJ and MvH. KVM contributed to the design and support of the quantum chemistry calculations and interpretation of the results. MKM and MSJ wrote the paper. All authors
edited and approved the manuscript.

*Competing interests.*  The University of Copenhagen (UCPH) has filed a patent application related to atmospheric iron chlorides on behalf of its inventors (MvH, MSJ). All other authors declare they have no competing interests.

*Acknowledgements.*  We thank Spark Climate Solutions for supporting this research as a part of the ISAMO project. Thanks to everyone on the ISAMO project for helpful comments and discussions.





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
