# Peer review of "Photocatalytic chloride to chlorine conversion by ionic iron in aqueous aerosols: A combined experimental, quantum chemical and chemical equilibrium model study"

_Aerosol Research, 2023_

## Author Response (AR1)

We thank all the referees for their careful analysis of our manuscript and the feedback they have provided. Here we describe the changes we have made to the manuscript in response to the reviews.

We have also provided a revised version where all changes have been highlighted.

**Replies to referee #1:**

Mikkelsen et al. study a very interesting and importing topic: The iron-catalyzed production of chlorine atoms. Unfortunately, the manuscript contains several errors. The main problems are:

1) It seems that the sulfate concentration in sea water (Table 1) is too large by a factor of 10. It should probably be 2.83E-2, not 2.83E-1. Please check! If this is the case, the results regarding sulfate in Fig. 5 are probably wrong.

- Thanks very much for this. Yes, the sulphate concentrations were too large by a factor of 10. This has been corrected and a discussion about low and high sulphate concentration has been added. All models showing "normal seawater", "high sulphate" and "high iron and high sulphate" concentration are included in the appendix.
- The discussion regarding the sulfate has now been changed. Please see the highlighted area in the attached pdf.

2) In Fig. 5, it appears as if [Fe(OH)2]+ is directly converted to [Fe(OH)4]- when the pH rises. I don't think this is correct. I expect that Fe(OH)3 is produced first, and it will precipitate out of the solution. Iron solubility is very low at high pH. Is this not considered by the AEM?

- In Figure 5, $[Fe(OH)_2]^+$ is not directly converted to $[Fe(OH)_4]^-$. $Fe(OH)_3$ has a very low concentration below 5 % due to its solubility, the species is shown in appendix B2. Yes these processes are considered in the AEM.

Once these problems are checked and fixed, I'd be ready to review a revised version of the manuscript.

Thanks very much for the feedback.

Additional comments:

1. Abstract, line 7: It is unclear what "title compounds from n=1..4" refers to. There is no chemical formula in the title, and n is undefined.

Thank you for the comment. The line has been changed to: "Here we present the results of experiments measuring the photolysis of $FeCl_3 * 6H_2O$ in specific wavelength bands, an analysis of the absorption spectra of $FeCl_n^{3-n}$ ($n = 1 .. 4$) made using density functional theory, and the results of an aqueous phase model that predicts the abundance of the iron chlorides with changes in pH and iron concentrations."

2. Line 31: "formation of sea spray aerosol [...] which are known to be ultrafine particles with high acidity". What is meant by "ultrafine"? Sea salt aerosol consists mainly of large (supermicrometer) particles. There is no high acidity when the sea salt aerosol is formed. Only after scavenging acids from the gas phase, a reduction of the pH is seen.

Thank you, this has now been changed to: "Traditionally, the tropospheric chlorine cycle begins with the formation of sea spray aerosol, which are known to be particles with high acidity."

3. Line 70: "Fe(III) and not Fe(OH)3" -> What is meant by this? The iron in Fe(OH)3 is also in the oxidation state III.

By line 70 is meant that the iron will not be in the form of $Fe(OH)_3$, but in the form of "free" Fe(III) due to the acidity of the solution.

4. The numbering of the reactions is inconsistent. It starts with (R1) and (R2) and then continues with (1) and (2). The letter "R" should be used for all reactions.

Thank you for noticing, this was a template mistake. All reactions are now numbered with R1, R2 etc.

5. Line 90: Why is reaction (4) listed when discussing photoreduction?  Iron is not reduced in reaction (4).

Correct. Line 85 states: "Iron(III) hydroxide complexes can be photolysed, similar to iron(III) chlorides, however only some of the iron is photoreduced;". Therefore, R5 and R6 are stated as photoreduction happens in R5 but not in R6.

6. Line 182: What is alpha? Please define it!

Thank you. This section has been rewritten. The revised text is as follows: "The temperature dependence of FeCl^{2+} and FeCl_2^{+} was modelled with the Seawater scenario of the AEM illustrated in Figure \ref{fig: temperature seawater} from 0 to 100 degrees C. The fraction of FeCl^{2+} increases with increasing temperature and the opposite trend is seen for FeCl_2^{+}. With varying temperatures, a change of 70 and 45 % is found for FeCl^{2+} and FeCl_2^{+}, respectively. Thus, temperature is an important parameter when calculating the rate of chlorine production from iron chlorides, as a change of 20 degrees C will significantly change iron speciation. "

7. Line 212 and elsewhere: Chloride is not a radical. Maybe this should be "chlorine radical" instead of "chloride radical"?

Thank you for noticing, this has been corrected throughout the article.

8. Tab. A1: Please define M, K and Delta H!

Thank you. The table text for Table A1 now reads: "Thermodynamic data for species added to the MinteQ models. The values were calculated from data by Tagirov et al. (2000). Molar weight (Mw ), equilibrium constant (log(K)), change in enthalpy (ΔH)"

References:

1. The citation of Gustafsson (2014) is incomplete. If there is no peer-reviewed publication, please provide at least a URL.

Thank you for noticing, a URL has been provided.

2. The DOI in the citation of Tomasi is malformed.

The DOI has been corrected.

3. The citation of Fenton (1894) should be listed under the letter "F", not "H".

The citation of Fenton is now listed under F.

**Replies to referee #2:**

The manuscript titled "Photocatalytic chloride to chlorine conversion by ionic iron in aqueous aerosols: A combined experimental, quantum chemical and chemical equilibrium model study" gives a good description about photocatalytic oxidation of chloride to chlorine based on iron (III) chloride complexes. The authors analyzed the formation constants of the iron chlorides under various conditions (i.e, different pH and chloride concentration) through combination of modelling, quantum chemical calculations and laboratory experiments. The authors found that the formation processes could be activated by sunlight especially for the sunlight at 440 nm. In summary, this manuscript is written well and would be benefit for the scientific community on atmospheric chloride and chlorine. However, the following comments should be considered.

Thank you for the feedback and please find our replies below.

"Line 7. "the title compounds from n=1..4" is not informative. What are the title compounds? What is "n=1..4"? "

- This has been corrected. This passage now reads:
  "Here we present the results of experiments measuring the photolysis of $FeCl_3 \cdot 6H_2O$ in specific wavelength bands, an analysis of the absorption spectra of $FeCl_n^{3-n}$($n = 1 .. 4$).."

"Line 31. Sea spray aerosols are known to be ultrafine particles with high acidity. The expression is not rigorous. The diameter of ultrafine particle is less than 50 nm (Guo et al. 2020). However, sea spray aerosols generally distribute in the range of 0.1–10 µm."

- This is corrected. The passage now reads: "Traditionally, the tropospheric chlorine cycle begins with the formation of sea spray aerosol, which are known to be particles with high acidity."

"Line 36-37. The authors showed the traditional chlorine cycle in the troposphere in Lines 30-35. Here I agree to the main argument that the chlorine production is poorly constrained. However, the logic of the text should be smooth."

- This is corrected. The passage now reads: "Chlorine production is poorly constrained and as a result, Cl is estimated to remove between 0.8 and 3.3 % of tropospheric methane, depending on the model."

"Why did the authors put Sect. 1.1 in the end of the Introduction? The author should show this section in certain line before line 62."

- Section 1.1 now starts at line 66 and is placed here because it is heavier on theory than the rest of the introduction.

"Please provide more description about sea water. Did the authors use real sea water or artificial sea water?"

- No seawater was used in the physical experiments. The seawater content used in the models are explained in Table 1. The seawater used in the quantum calculations is described by the relative permittivity of seawater stated in Table 2.

"Table 1. "9.76·10−13" "13 ×10−4". What are the differences between "·" and "×"."

- This is a mistake in format. All are corrected to "×".

"Line 116-117. These two lines may be not necessary because it is very clear from the subtitles (2.1, 2.2, and 2.3)."

- Thank you for noticing. These two lines are added for a simple overview of the section, for readers that have not read all the subtitles.

"Please double check the expression about Cl, Cl·, Chloride."

- Thank you we have checked and have changed in numerous examples please see the 'track changes' version of the manuscript.

"The temperature dependences of the parameter α about two iron (III) chloride complexes are shown but the note of parameter α was missing in this manuscript."

- Thank you, this has been corrected and the parameter alpha has now been changed. Please see the 'track changes' version of the manuscript.

"What does the error bars represent? Standard deviation?"

- Yes they do represent standard deviation, this is now described in the article. This passage is added in the figure text of Figure 9: "The error bars represent the standard deviation."

"The conclusion part is tedious. In addition, some expressions such as "as shown by Li et al.", "based on the results" should be placed in the Result and Discussion."

- Thank you for the opportunity to improve the presentation of our work. We have rewritten the conclusion. Please see the 'track changes' version of the manuscript.

**Replies to referee #3:**

This is an interesting laboratory study on the conversion of chloride ions in aqueous solution into molecular chlorine, catalysed by iron (III) ions. The paper is generally well-written and illustrated, and describes careful experimental work. The results are potentially important for the atmospheric chemistry of the marine boundary layer. I have mostly the following minor revisions to suggest:

Thank you for your review and helpful suggestions.

**Abstract:**

"line 2: change to "Prior aerosol chamber experiments" to make it clear that this is not work in the present study."

- Thank you, done.

"...experiments to measure the photolysis of FeCl3.6H2O..."

- Thank you, done.

"line 7: specify FeCln(3-n)+ for n=1-4, rather than "title compounds"

- Thank you, done.

"line 14: suggest putting: "chlorine (Cl2/Cl) from chloride (Cl−)" at the beginning of the abstract, since "chlorine" and "chloride" are referred to throughout the abstract."

- Thank you, done.

**Main body:**

"line 25: suggest rephrasing to something like "Iron from the Sahara dust is released and forms iron chlorides with chloride from the sea spray. Iron chlorides can absorb sunlight, releasing a chlorine atom. The chlorine is emitted from the aerosol as molecular chlorine (Cl2), which is then photolysed by sunlight to yield atomic Cl in the gas phase.""

- Thank you, done.

"line 59: change "This work" to "That study...""

- Thank you, done.

line 61: do you mean "chloride", rather than "chlorine"?

- Thank you, changed.

line 63: these chlorine species are a different list from those discussed in the previous paragraph. Perhaps you should point this out?

- Thank you, done.

line 71: change to "Given the presence of chloride and ..."

- Thank you, done.

line 82: change to "At higher pH the formation of iron(III) chloride complexes competes with iron hydroxide complexes"

- Thank you, done.

line 87: the sentence "The UV absorption spectra of Fe(OH)2+ and Fe(OH)2+ have been measured" ends abruptly - what is the point of saying this?

- This have been changed, please see the 'track changes' version of the manuscript.

line 98: how will OH enhance Cl2 production? Are reactions 8 - 10 those proposed by Wittmer and Zetzsch (2017)?

- Thank you, this should be clearer. As OH enhance the production of Cl radicals in reaction R10 to R12 (In the revised version), this Cl radicals will then form $Cl_2$ (reaction R13 to R15). The text

now reads: "Wittmer and Zetzsch (2017) proposed that the production of OH will enhance $Cl_2$ production due to the production of a chlorine radical, shown in reactions R10 to R12. "

line 104: specify which reactions are being referred to, by their numbers

- Thank you, this have been changed in the report. please see the 'track changes' version of the manuscript.

line 112: "Once in the gas phase, molecular chlorine ..."

- Thank you, changed.

page 6, caption of Table 1. Are the "Species and concentrations listed for each AEM scenario." the initial or modelled concentrations?

- It is the initial, it has been changed in the article

line 116: Explain briefly what each part is for e.g. AEM to predict the concentrations of FeCln3-n species, ab initio calculations to predict absorption rates, and lab experiments to prove Cl is produced as predicted by models/measure Cl reaction with methane/show Cl reaction is the major loss process for CH4.

- Thank you, done.

line 123: was FeCl4- already present, or added automatically?

- Iron(III) tetrachloride could not be added because the thermodynamic data is not available. The passage now reads: "The species $FeCl^{2+}$, $FeCl_2^{+}$, and $FeCl_3$ were manually added to the database, shown in the Appendix Table A1, however, $FeCl_4^{-}$ could not be added because the thermodynamic data is not available."

129: "...as a function of pH in section 3.1."

- Thank you, done.

page 7, Figure 2 shows $\epsilon$, but the caption states $\epsilon r$

- Thank you, changed.

line 137: does this mean the ab initio predictions for FeCl4- are not reliable?

- No, only that the accuracy could potentially be increased.

page 8, Figure 3: consider renaming some items on the diagram to make it clearer: Chamber 1 = sample chamber, Lamp 1 = Xenon lamp, Chamber 2 = Photolysis chamber, Lamp 2 = UV LED, Active Carbon = active carbon trap

- It has now been done.

page 8, Table 2: Are the $\epsilon r$ for NaCl solutions for different conditions? Temperature, pH and other parameters discussed in different places? Why are so many (similar) values used? In the two last lines, is "solid FeCl3" anhydrous FeCl3 or FeCl3.6H2O?

- The $\epsilon_r$ for NaCl solutions are based on Midi et al. (2014) where it is seen that a small change in the relative permittivity changes the conductivity and is dependent on the frequency. These values were initially used, as we had a hypothesis that it would noticeably perturb the UV-VIS spectrum. However, it did not and instead the arbitrary investigation was done with a broader step between the values. The "solid $FeCl_3$" are estimations on the solid relative permittivities of $FeCl_3$ anhydrous as these values could not be found in the literature.

line 141: $\sigma_A$ in equation 15

line 142: $\varphi_A$ in equation 15

- $\sigma_A$ and $\varphi_A$ are fixed

line 146: change to "...Figure 3. Gases are introduced in a 4-port valve ("4V"), which was used to choose a flow-through or loop pattern."

- Thank you, done.

line 147: the experimental description is written in the present tense e.g. "A sample of FeCl3·6H2O is placed in Chamber 1 ...", whereas the rest of the paper is written in the past tense.

- Corrected

line 150: if the cold trap and active carbon trap remove organics and carbon dioxide, how do the instruments measure the methane and CO2? What does each of the instruments measure? Why are both needed?

- The Picarro is relied on for more accurate measurements of $CH_4$ and $CO_2$. We want to measure $CO_2$, because we want a pure $N_2$ atmosphere and if we measure any $CO_2$ it is an indication of an impure atmosphere. The Axetris is used for a live measurement, it was mainly used for the introduction of methane as is has a very fast measuring capacity. In contrast the Picarro takes longer to measure, but measures more accurately. The cold trap does not remove $CH_4$, only large organic molecules.

line 155: "In each experiment" is unclear. Do you mean at the start of each experiment? For each experiment? Before each experiment?

- This has been changed to: "At the beginning of every experiment..."

line 155: "FeCl3.6H2O"

- Thank you, done.

line 159: "...measurements are taken to verify system stability."

- Thank you, done.

page 10, Figure 5: consider swapping the order of species in the legend to read naturally from left to right in the order they appear in the plots (low pH to high pH)? e.g. chlorides (red), Fe2+(yellow), sulphates (green), hydroxides (blue), Fe(OH)3- (purple). Also, specify the temperature in the figure caption.

- The temperature is now stated in the caption. The order of species have not been changed.

line 170: "...as a function of pH for the seawater iron concentration of 9.76x10-13 mol/kg."

- Thank you, done.

line 171: "Above this, the dominant ..."

- Thank you, done.

line 172: you have already said the results are in Figure 5, so saying compare with Figure 5 doesn't make sense

- Changed to "in Figure 5"

line 173: in the Introduction (line 59) you state that FeCl2+ doesn't form, so why does the model think it is one of the dominant species?

- Line 59: "This work indicates that FeCl^(2+) will not be formed...". The model states that FeCl2^(+) is one of the dominant species. It's two different compounds.

line 179: Figure 5 shows the complex ions, not functional groups/moieties.

- Changed sentence

line 182: alpha does not seem to be defined? I assume it is fraction of species and presumably not the same alpha as from equation A3?

- This section has been rewritten. The revised text is as follows: "The temperature dependence of FeCl^{2+} and FeCl_2^{+} was modelled with the Seawater scenario of the AEM illustrated in Figure \ref{fig: temperature seawater} from 0 to 100 degrees C. The fraction of FeCl^{2+} increases with increasing temperature and the opposite trend is seen for FeCl_2^{+}. With varying temperatures, a change of 70 and 45 % is found for FeCl^{2+} and FeCl_2^{+}, respectively. Thus, temperature is an important parameter when calculating the rate of chlorine production from iron chlorides, as a change of 20 degrees C will significantly change iron speciation. "

line 184/5: Say what temperature figure 5 is illustrating.

- Thank you, done.

Caption to Figure 6: why include "(FeS)" here and not in Figure 5?

- Thank you, we have now made this change to Figure 5.

line 191: Why only for FeCl3? Why not include calculated UV-Vis spectra for all species?

- All permittivities were calculated for FeCl$_3$ as part of the initial calculations. We found that only calculating 1.0, 2.0, 69.0 and 78.4 gives a complete picture of the response of the chromophore to changes in the permittivity, just these were calculated for the other compounds.

line 191: rewrite to something like "...Appendix figure B1. Spectra show an increasing red shift..." (the red shift does not result from the spectra being in appendix figure B1!).

- Thank you, done.

Figure 7. The y axis label is "Absorption Rate 10-3" - is this 10-3 times the absorption rate (i.e. absorption rate ~103 s-1 nm-1), or is a factor of 10-3 factored out (i.e. absorption rate ~10-3 s-1 nm-1)?

- the $10^{-3}$ is factored out, so numbers range from 0.0 x $10^{-3}$ to 4.0 x $10^{-3}$.

line 213: "surface" here refers to the Earth's surface, not a transition state energy surface?

- Yes, we have changed the sentence: "Near the Earth's surface..."

line 214: replace "inconsequential" with "negligible"

- Thank you, done.

line 220: "for the dissociation of FeCl2+" - and presumably FeCl3 and FeCl4-?

- This section has now been changed. The sentence now says: "Sunlight at the surface has photons with enough energy for the dissociation of FeCl_2^+, FeCl_3, and FeCl_4^-. "

line 225: It's not that different from the other values compared to the difference between e.g. solid and seawater/water for FeCl2+, FeCl3 and FeCl4-? I don't know that it is enough to clearly show it is an outlier? Surely the fact that they differ is enough to argue for considering solvent effects, it doesn't even need to be a outlier?

- We agree with your statement that it doesn't need to be an outlier for the discussion to be validated. However, as most quantum chemical calculations are done in vacuum, we find it important to state that vacuum does not necessarily closely relate to results calculated in solvents. And therefore we state vacuum as a special case.

line 228: "integrated absorption rates are listed in Table 3". Is this just figure 7 integrated? Or is it multiplied by something?

- Yes, and all numbers are multiplied by 10^-3.

line 232: "in Table 3, FeCl2+..."

- Thank you, done.

line 240: "which increases the importance of this species." By how much?

- Increases the importance to us, in the sense that it could be an important/central compound in the mechanism discussed in the introduction.

line 249: Presumably you mean the reaction CH4 + Cl -> CH3 + HCl? This should be specified either here or in the previous section.

- This has now been written out in the text.

line 252: Explain why the normalisation is done, and why it is important.

- This has now been described, the revised text now states: "As the bandpass filters do not have the same wavelength width or energy throughput, Figure 4, the energy normalization contradicts this effect."

line 255: Presumably the average of the three repeats at each bandpass? But this paragraph is the only time it is referred to as an average.

- Yes, it is the average of the three repeats

Table 3. I think these rates are missing a factor of 10-3. Assuming this is an integration of the lines from figure 7, for FeCl2+in a vacuum, approx area of triangle under the graph (ignoring the 10-3 in the y axis label) = 1/2 * base* height = 0.5*350*0.4 =50 s-1. This is about the same as the value (to within triangle assumptions!), but it uses a y value of 0.4 rather than 0.4x103 or 0.4x10-3 (depending on what "Absorption rate 10-3" is supposed to mean), so shouldn't the rate be 1000 x bigger/smaller than this? (It presumably shouldn't affect the % deviation though).

- That is completely right. All numbers are multiplied by $10^{-3}$. This has been stated now.

line 263: the Cl concentration doesn't appear to be reported anywhere.

- This calculation is detailed in the Appendix A6, as stated in line 263-265.

line 264: The equation is shown in A6, but no calculations or results are included.

- These are straightforward calculations so we described the equations and method, but decided not to present the numbers. We would of course make them available if anyone is interested.

line 266: "...the measured change in the delta value ..."

- Thank you, done.

line 270: the observed values are always higher (except with no bandpass), though they are within error, but it seems systematic.

- We agree, the values are within the mutual error bars, and yes there does seem to be a small systematic error. Due to some technical details we were not able to complete an intercalibration of our (new) instrument during the study period and so have no choice but to live with the data as it is.

line 272: Wouldn't a calibration offset affect all bandpasses?

- Yes but please see previous comment.

Figure 9 caption: y axis label is Δδ13CH4 , not Δδ13C −CH4

- The Picarro does not measure $\delta^{13}C$, but only $\delta^{13}CH_4$. We have therefore decided to state it like this.

line 277: [Cl-] is the same across simple, sulphate, and seawater AEM scenarios, so how have [Cl-] conditions been varied?

- No they have not been varied, this must be a misunderstanding. The passage not states: "Modelling, quantum chemical calculations and laboratory experiments have provided crucial insight into how iron(III) chloride chromophores behave in aqueous solutions under varying conditions of pH."

line 280: "addition of sulphate decreased the abundance"

- Thank you, done.

line 290: "Slowest rate is 0 s-1" is not a very informative result? Better to say FeCl2+ does not absorb sunlight under atmospheric conditions? You could include lowest (non-zero) integrated absorption rates (FeCl2+ with 131/134 s-1 in seawater/water, respectively, or FeCl4- in a vacuum (113 s-1), though vacuum conditions are obviously not going to be valid in the atmosphere).

- This section has been changed to: "For $FeCl^{2+}$, $FeCl_2^{+}$, and $FeCl_4^{-}$ inclusion of the energy threshold significantly decreased the absorption rates. $FeCl_3$ was the only species without sunlight-limiting energy thresholds in any solvent. In water, $FeCl_3$ has the fastest integrated absorption rate of $259 * 10^{-3}$ s$^{-1}$ and $FeCl^{2+}$ had the overall slowest rate."

line 297: I assume this is referring to maximum CH4 removal from the 436 nm bandpass filter, and the maximum predicted absorption rate was at ~425 nm for FeCl3 in water/seawater? But this does not seem to be stated in the body of the paper, nor that this was the point of the experiments.

- This conclusion has been changed, please see the revised version.

line 322: "FeCl3.6H2O"

- Thank you, done.

line 326: you state here that the sample was heated. It has been reported that heating FeCl3.6H2O can produce HCl, so the sample may not necessarily have remained as FeCl3 if it has been heated before the experiment took place. Was there any evidence of HCl production?

- There was no evidence of anything happening during the heating of the chamber. However, we did not measure HCl directly with any instrument

line 337: Is leak rate the rate of loss of methane before the lamps are turned on, or are other gases also monitored? If it is just methane, is this affected by heating the sample?

- Yes the leak rate is the loss of methane in the system due to leaks in the system. This is measured before the lamps are turned on and after the lamps are turned off. This is not affected by heating the sample.

line 339: "FeCl3.6H2O"

- Thank you, done.

line 339: "are discussed in methods (Section 2.3)".

- Thank you, done.

Figure A2. Why is reflected spectrum included in the figure, but never referred to?

- This spectrum was recorded simultaneously and simply gives some additional confirmation of the result.

Figure A3 caption. These are not "results"!

- Thank you, this has been changed.

Figure B2. What about FeCl3 or FeCl4-, which are not included? Or does the model predict negligible amounts of them? In which case, there would be very little Cl production from FeCl3 despite its predicted integrated absorption rate?

- They are included in the model, as described, however the model predicts infinitesimal concentrations of them in the aqueous phase.